# Fully Human Herpesvirus-Specific Neutralizing IgG Antibodies Generated by EBV Immortalization of Splenocytes-Derived from Immunized Humanized Mice

**DOI:** 10.3390/cells13010020

**Published:** 2023-12-21

**Authors:** Sebastian J. Theobald, Elena Fiestas, Andreas Schneider, Benjamin Ostermann, Simon Danisch, Constantin von Kaisenberg, Jan Rybniker, Wolfgang Hammerschmidt, Reinhard Zeidler, Renata Stripecke

**Affiliations:** 1Department I of Internal Medicine, University Hospital Cologne, Faculty of Medicine, University of Cologne, 50937 Cologne, Germany; jan.rybniker@uk-koeln.de (J.R.); renata.stripecke@uk-koeln.de (R.S.); 2Center for Molecular Medicine Cologne (CMMC), University Hospital Cologne, Faculty of Medicine, University of Cologne, 50937 Cologne, Germany; 3German Center for Infection Research (DZIF), Partner Site Bonn-Cologne, 50931 Cologne, Germany; 4Clinic of Hematology, Hemostasis, Oncology and Stem Cell Transplantation, Hannover Medical School, 30625 Hannover, Germany; schneider.andreas@mh-hannover.de (A.S.); danisch.simon@mh-hannover.de (S.D.); 5German Center for Infection Research (DZIF), Partner Site Hannover-Braunschweig, 30559 Hannover, Germany; 6Research Unit Gene Vectors, Helmholtz Center Munich, German Research Center for Environmental Health, 81377 Munich, Germanyhammerschmidt@helmholtz-muenchen.de (W.H.); 7German Center for Infection Research (DZIF), Partner Site Munich, 81377 Munich, Germany; reinhard.zeidler@helmholtz-munich.de; 8Institute of Structural Biology, Helmholtz Center Munich, German Research Center for Environmental Health, 81377 Munich, Germany; 9Department of Obstetrics, Gynecology and Reproductive Medicine, Hannover Medical School, 30625 Hannover, Germany; vonkaisenberg.constantin@mh-hannover.de; 10Department of Otorhinolaryngology, Munich University Hospital, 81377 Munich, Germany; 11Institute of Translational Immuno-Oncology, University Hospital Cologne, Faculty of Medicine, University of Cologne, 50937 Cologne, Germany; 12Cancer Research Center Cologne Essen, University Hospital Cologne, Faculty of Medicine, University of Cologne, 50937 Cologne, Germany

**Keywords:** HCMV, EBV, herpesvirus, humanized mice, antibodies, immortalization, dendritic cells, virus-like particles

## Abstract

Antiviral neutralizing antibodies (nAbs) are commonly derived from B cells developed in immunized or infected animals and humans. Fully human antibodies are preferred for clinical use as they are potentially less immunogenic. However, the function of B cells varies depending on their homing pattern and an additional hurdle for antibody discovery in humans is the source of human tissues with an immunological microenvironment. Here, we show an efficient method to pharm human antibodies using immortalized B cells recovered from Nod.Rag.Gamma (NRG) mice reconstituting the human immune system (HIS). Humanized HIS mice were immunized either with autologous engineered dendritic cells expressing the human cytomegalovirus gB envelope protein (HCMV-gB) or with Epstein–Barr virus-like particles (EB-VLP). Human B cells recovered from spleen of HIS mice were efficiently immortalized with EBV in vitro. We show that these immortalized B cells secreted human IgGs with neutralization capacities against prototypic HCMV-gB and EBV-gp350. Taken together, we show that HIS mice can be successfully used for the generation and pharming fully human IgGs. This technology can be further explored to generate antibodies against emerging infections for diagnostic or therapeutic purposes.

## 1. Introduction

Immunoglobulins (Ig) can contribute to resolve infections by inducing immune effector functions like antibody-dependent cell-mediated cytotoxicity (ADCC) and antibody-dependent cellular phagocytosis (ADCP), and directly by blocking virus spreading within the infected individual. Virus-specific neutralizing antibodies (nAbs) represent an attractive option for the treatment and prevention of acute viral infections or infections that are drug resistant or recurrent. For example, during the recent SARS-CoV-2 pandemic, antibodies against the viral Spike protein with neutralizing properties were developed and successfully used to treat patients [1], and Palivizumab (Synagis^®^) efficiently protects from infections with the respiratory syncytial virus (RSV). Also, pooled hyper-immunoglobulin-preparations (HIG) containing HCMV-specific IgGs at a high concentration are in clinical use [2], but their composition and efficacy varies from batch to batch.

The Human Cytomegalovirus (HCMV) and Epstein–Barr virus (EBV) are almost ubiquitous herpesviruses [3,4], which are controlled by the host’s immune system [5,6] so that they normally remain silent in a latent state of infection, characterized by the expression of only a minimal set of viral genes [7]. However, both viruses are significant human pathogens causing various severe diseases in healthy individuals and particularly in immunocompromised patients. For example, HCMV reactivation in patients regularly causes pneumonia, colitis or retinitis, and is associated with low survival [8,9,10]. Furthermore, congenital or neonatal HCMV infection of newborns is associated with serious complications like mental retardation, seizures and hearing loss [11]. Primary EBV infection regularly causes infectious mononucleosis (IM), linked with increased risks of, e.g., Hodgkin lymphoma (HL) and multiple sclerosis (MS) later in life [12]. Further, EBV is a well-known oncogenic virus that contributes to several types of malignancies, including Hodgkin lymphoma (HL), Burkitt lymphoma (BL), nasopharyngeal carcinoma (NPC), gastric cancer (GC) [13] and post-transplant lymphoproliferative disorder (PTLD) [14].

Despite their clinical relevance, HCMV- or EBV-specific therapeutic antibodies have not been marketed yet. Therefore, novel approaches to generate high affinity and highly neutralizing antibodies could aid to move the field forward.

In our laboratory, we have employed NRG mice, a radiation-resistant immuno deficient mouse strain, that, in our hands, showed more consistent human immune system (HIS) reconstitution after humanization with cord blood CD34^+^ hematopoietic stem compared with other strains. NRG HIS-mice immunized with autologous induced dendritic cells (iDCs) expressing the HCMV gB protein developed human class-switched memory B cells secreting immunoglobulins of the IgG subtype [15]. We were able to isolate single B cells and clone human gB-specific and virus-neutralizing monoclonal IgGs [15]. More recently, we were able to show that the immunization of humanized mice with lentiviral vectors (LVs) expressing HCMV-gB also accelerated B cell development [16].

Despite these achievements, the B cells obtained from humanized mice die rapidly in vitro, and, therefore, their supply is limited, e.g., for downstream in vitro affinity maturation approaches. In the past, it has been demonstrated that immunization of humanized mice with iDCgB resulted in higher antigen-specific B cell responses than HCMV infection [15]. Therefore, we used, in this current study, iDCgB and EBV virus-like-particle (VLP) vaccines as immunization strategy. Compared to infection models, immunization induces a narrow antigen-specific immune response. Here, we demonstrate that B cells recovered from immunized humanized mice can be immortalized with EBV using a feeder cell system. Further, we screened immortalized B cell lines (LCLs, lymphoblastoid cell line) secreting IgGs and show that some of the secreted antibodies show antigen binding and neutralization capacity. These results open a new approach for generation and discovery of novel fully human IgG antibodies for pharmacologic uses.

## 2. Materials and Methods

### 2.1. Ethics Statements

Collection of umbilical cord blood (CB) was performed at the Department of Gynecology and Obstetrics (Hannover Medical School) with study protocols approved by the Ethics Review Board (approval Nr. 4837) and after informed consent of the mothers. All experiments involving mice were approved by the Lower Saxony Office for Consumer Protection and Food Safety—LAVES (permit number: 16/2222) and performed in accordance with the German animal welfare act and the EU-directive 2010/63.

### 2.2. Cell Lines and Primary Cells

HEK-293T cells (human embryonic kidney cells, ATCC Manassas, VA, USA) and MRC-5 (human fetal lung fibroblasts, American Type Culture Collection, ATCC) were expanded and cultured in Dulbecco’s Modified Eagle Medium (DMEM; ThermoFisher, Waltham, MA, USA) containing non-essential amino acids (ThermoFisher), 10% fetal bovine heat-inactivated serum (FBS, fetal bovine serum, HyClone, Logan, UT) and 1% penicillin/streptomycin (Merck Millipore, Billerica, MA, USA) at 37 °C with 5% CO_2_. PCI-1 (human laryngeal squamous cell carcinoma, a kind gift of Prof. Theresa Whiteside, Pittsburgh, PA, USA), Raji (human EBV-positive Burkitt lymphoma, ATCC), and LL8 (mouse fibroblast expressing CD40-ligand, kindly provided by Dr. Andreas Moosmann, LMU Munich) cell lines (referenced in [17]) were expanded and cultured in Roswell Park Memorial Institute medium (RPMI 1640, Gibco, Thermo Fisher, Waltham, MA, USA) containing 8% FBS (Bio & SELL, Nurnberg, Germany) and 1% penicillin/streptomycin (Gibco, Thermo Fisher, Waltham, MA, USA) at 37 °C with 5% CO_2_. LEXL5 is a conditionally immortalized B cell line (kindly provided by A. Moosmann) generated by co-culture with LL8 cells providing CD40L stimulation and in the presence of 2 ng/mL interleukin (IL)-4 (2 ng/mL, R&D Systems, Minneapolis, MN), 10% FBS (Bio & SELL, Nurnberg, Germany) and 1% penicillin/streptomycin (Gibco) at 37 °C with 5% CO_2_ [18]. Peripheral blood mononuclear cells (PBMCs) derived from CB-units were isolated by Ficoll gradient centrifugation. CD14^+^ and CD34^+^ cells were isolated by immune magnetic beads following the vendor’s instructions (Miltenyi Biotech, Bergisch-Gladbach, Germany). Magnetic isolation of highly pure CD34^+^ cells for humanizing mice was performed twice, as we described before [19]. A stable PCI-1 clone that expresses gp350 was obtained by transfection with a suitable expression plasmid followed by selection with G418.

### 2.3. Production Lentiviral Vectors and Generation of iDCs Expressing HCMV-gB

Lentiviral vectors and iDCgB were produced as previously described [15]. Briefly, a third-generation lentiviral vector expressing full-length HCMV-gB and a bicistronic vector expressing human granulocyte-macrophage colony-stimulating factor (GM-CSF) and interferon-α (IFN-α) were produced in HEK293T cells and concentrated by ultracentrifugation. CB-CD14^+^ monocytes, autologous to the CD34^+^ cells used to humanize mice, were were used and co-transduction was performed with the lentiviral vectors to generate iDCgB (LV-GM-CSF/IFN-α at 2.5 mg/mL p24 equivalent and LV-HCMVgB at 3.0 mg/mL p24 equivalent for transduction of 2.5 × 10^6^ cells) as described [15]. Quality control was performed by flow cytometry detection of DC maturation markers (HLA-DR, CD80, CD86) and of the HCMV-gB antigen. For long-term storage, iDCgB cells were cryopreserved. For immunizations, cells were thawed, washed, re-counted and administered as previously described [15].

### 2.4. Production of Epstein–Barr Virus-like Particles Containing gp350

EBV-like particles (EB-VLPs) were produced in HEK-293 cells stably transfected with a maxi-EBV plasmid (kindly provided by Prof. Wolfgang Hammerschmidt, Helmholtz Munich, see also [20]). These VLPs do not contain many of EBVs microRNA (miRNA) and the VLP Cre5 virus producer cell clone 6507 was used. The production of VLPs was induced after transfection with plasmids expressing BZLF1 and BALF4 (for details see [21,22]). The cell supernatant containing the EB-VLPs was collected three days after transfection, centrifuged at 300× *g* for 10 min and then at 2000× *g* for 20 min. The supernatants were filtrated through a 0.8 µm filter. The cell-free VLP suspensions were stored at 4 °C.

### 2.5. HCT and Immunization of NRG Mice

Female NOD/Rag1^null^/IL2Rγ^null^ (NRG) mice were obtained from The Jackson Laboratory (JAX, Bar Harbor, ME, USA) and bred in-house under pathogen-free conditions. Mice were transplanted with CB CD34^+^ cells as previously described [19]. Briefly, 5 weeks old mice were irradiated (450 cGy) using a [^137^Cs] column irradiator (Gammacell 3000 Elan; Best Theratronics, Ottawa, ON, Canada) and 4 h later transplanted with 2.0 × 10^5^ human CB-CD34^+^ cells by intravenous (i.v.) injection. To generate humoral responses against HCMV-gB, humanized mice were immunized at 6, 7, 10 and 11 weeks after HCT subcutaneously (s.c., near the anatomical regions of the inguinal and axillary lymph nodes) with 5.0 × 10^5^ iDCgB cells. To stimulate immune responses against of EBV, humanized mice were immunized at 10, 11, 14, 16 and 18 weeks after HCT with 5 × 10^6^ EB-VLPs injected i.v.

### 2.6. Flow Cytometry

Splenocytes obtained from humanized mice were dissociated and incubated with a hypotonic solution (0.83% ammonium chloride/20 mM HEPES, pH 7.2, for 5 min at RT) to lyse the erythrocytes. Immune staining of the B cells was performed essentially as described [15] after a blocking step and staining with fluorochrome-labeled antibodies. Anti-human CD45-Pacific blue (Biolegend, 304022); p27-287 (gB) (kindly provided by Prof. Michael Mach, Erlangen); Anti-mouse IgG-Alexa647 (Biolegend, 405322); Anti-mouse IgG-Alexa488 (Biolegend, 405319); p63-27 (iE1) (kindly provided by Prof. Michael Mach, Erlangen); anti-mouse-IgG-Alexa647 (Biolegend; 405322); anti-human IgG -Alexa647 (Jackson ImmunoResearch, 109-605-003). Flow cytometry acquisition was performed with a LSR II device (BD Biosciences, Heidelberg, Germany). Data analysis was performed with FlowJo (Treestar Inc., Ashland, OR, USA).

For the detection of gp350-specific human IgG antibodies, parental PCI-1 cells and PCI-1 cells stably expressing gp350 were incubated with supernatants from LCLs, followed by incubation with a anti human-IgG Alexa647-labeled secondary antibody.

### 2.7. Immortalization of B cells after EBV Infection and Expansion

Splenocytes obtained from humanized mice were dissociated into single cell suspensions and used immediately. Infection was performed with the EBV virus strain B95.8 at a multiplicity of infection (MOI) of 0.1. 5 × 10^6^ splenocytes were incubated for 3 h at 37 °C and 5% CO_2_ with EBV in RPMI 1640 media (Gibco, Thermo Fisher, Waltham, MA, USA), supplemented with FBS (described above), Cyclosporine A (CsA, 1µg/mL, Novartis, Basel, Switzerland) and IL-4 (2 ng/mL R&D Systems, Minneapolis, MN) and 1% penicillin/streptomycin (Gibco, Thermo Fisher, Waltham, MA, USA). Irradiated feeder cells prepared on the day prior to the EBV infection were used to harness immortalization. LL8 and LEXL5 cells were irradiated at 180 Gy in a ^137^Cs device. 1 × 10^6^ irradiated LL8 cells were seeded per well on 96-well cell culture plates. On the next day, 2.5 × 10^4^ infected cells were added per well. Then, 2.5 × 10^4^ irradiated LEXL5 cells were added per well of the cell mixture. Half of the media was replaced with newly prepared media weekly. CsA was added during the first 4 weeks of culture, to deplete the T cells from the culture. IL-4 was kept during the whole culture to improve the B cell viability and expansion.

### 2.8. ELISA for Detection of IgGs Binding to gB or to gp350

Detection of Igs secreted by the immortalized B cells and binding to HCMV gB-binding IgGs was performed by ELISA as previously described [15]. Briefly, recombinant gB protein (kindly provided by Dr. Marija Backovic, Institute Pasteur, Paris, France) was used for coating. The wells were washed to remove excess protein and blocked with 5% FBS in phosphate-buffered saline (PBS) at 37 °C for 2 h. Next, plates were incubated with undiluted LCL supernatants at 37 °C for 2 h. After additional washing, detection was performed with anti-human IgG or IgM or IgA antibodies conjugated with horseradish peroxidase (HRP) (1:1000 in PBS/2% FBS) and developed with TMB. The OD 450 was measured with an ELISA plate reader (Molecular Devices). ELISA to detect Igs binding to EBV-gp350 was performed following a similar procedure, using recombinant gp350-His protein (Sino Biological. Beijing, China) for coating.

### 2.9. Flow Cytometry Assay to Detect gB Binding IgG

A clonal surface gB-expressing HEK-293T cell line (previously described in [23]) was used to determine gB binding of LCL supernatants. Therefore, supernatants were incubated with the gB-expressing cell line for 1 h at 4 °C. Next, cells were washed and incubated with an anti-IgG-Alexa647 antibody (Biolegend) to determine antigen binding. Only secondary antibody was used as a control.

### 2.10. EBV In Vitro Neutralization Assay

Recombinant EBV expressing GFP [24] was provided by Prof. Hammerschmidt. EBV/GFP was pre-incubated with different volumes (20, 50, 100, 150 and 200 µL) of LCL supernatants at 37 °C for 30 min and then added to 1 × 10^5^ Raji cells. Cells were cultivated for 48 h and afterwards analyzed by flow cytometry for GFP-positive cells, indicative for EBV cell infection. GFP^+^ Raji cells incubated with the supernatant of control LCLs (derived from non-immunized mice) and infected with EBV/GFP were used for normalization.

### 2.11. HCMV In Vitro Neutralization Assay

MRC-5 fibroblasts were seeded onto 96-well plates at a density of 0.5 × 10^5^ cells per well. On the next day, MRC-5 cells were incubated with LCL supernatants and HCMV-gLUC, MOI 0.1 for 4 h at 37 °C. Next, cells were washed twice with PBS and fixed for 15 min with ice-cold ethanol. After one additional washing step, cells were incubated with purified p63-27 (anti-IE1 antibody, kindly provided by Prof. Michael Mach, Erlangen, Germany) for 1 h at 37 °C. Cells were washed thrice and incubated with an anti-mouse IgG-Alexa647 (Biolegend, 1:1000) for 45 min at 37 °C. After three washes with PBS, cells were stained with DAPI (DAKO) for 10 min and imaged with a fluorescent microscope. Percent IE-1 positive cells were calculated on total DAPI positive cells.

### 2.12. Statistical Analysis

Statistical analyses were carried out using GraphPad Prism V7.0 software (GraphPad Software, La Jolla, CA, USA). All statistical methods used and standard deviations are mentioned in the respective figure legend. In summary, we used *t*-test with Welsh`s correction and One-way-ANOVA depending on the comparison.

## 3. Results

### 3.1. Immunizations of HIS NRG Mice for Pharming Human IgG Antibodies against HCMV-gB or EBV-gp350

We have previously shown that HCMV infection [25] or iDCs [15], or a LV expressing HCMV-gB [16] stimulated the production of Igs in humanized NRG mice. In the current studies, we evaluated if B cells derived from HIS mice immunized either with iDCgB or with EB-VLP could be immortalized in vitro. NRG mice were transplanted with CB-derived CD34^+^ stem cells and iDCgB s.c. immunizations were carried out at weeks 6, 7, 10 and 11 post-HCT as previously reported [15] (Figure 1A). Alternatively, humanized NRG mice were immunized with EB-VLPs at 10, 11, 14, 16 and 18 weeks post-HCT. For both procedures mice were sacrificed at week 25 post-HCT. The frequencies of human CD45^+^ cells in the spleens were comparable between all experimental groups with an average of 60% to 70% (Figure 1B). Plasma from HIS-mice immunized with iDCgB and EB-VLP were tested for IgG antigen specificity. ELISA assays performed with plasma confirmed the presence of human IgGs specific against HCMV-gB (Figure 1C) and EBV-gp350 (Figure 1D), respectively. The levels of detectable IgGs were significantly higher in mice immunized with iDCgB (*p* = 0.0239) or with EB-VLP (*p* = 0.0334) than in non-immunized control mice. These results demonstrated that immunized HIS-mice mounted antigen-specific humoral responses.

### 3.2. Immortalization of B Cells Derived from Humanized Mice

Fresh B cells recovered from mice were used for immortalizations. Therefore, splenocytes were isolated and the single cell solution was infected with EBV for 3 h. Afterwards, splenocytes were co-cultured with irradiated feeder cells (namely LL8 and LEXL5), which provide a favorable microenvironment for an efficient immortalization (Figure 2A) [18]. IL-4 was added to the cell culture weekly for 4 weeks to improve the recovery of immortalized B cells (Figure 2A).

### 3.3. Screening of Immortalized B Cells for Production of Secreted or Membrane Bound Igs Reactive against gB

Cell culture supernatants of immortalized B cell lines derived from iDCgB immunized mice were screened for the presence of gB-specific antibodies by ELISA. As shown in Figure 2B, approximately 5% of the generated lines produced higher levels of IgM binding gB. For IgG the frequency was approximately 10%, while only a few of the cell lines secreted IgA. In order to verify the specificity of IgG-producing lines, we employed a second flow cytometry-based assay using 239T cells stably transduced with a HCMV-gB expression plasmid and therefore expressing HCMV-gB on the cell surface [23]. Importantly, ELISA positive supernatants were also positive in our confirmatory FACS assay, indicated by a significantly (*p* = 0.0107) higher signal compared to supernatants derived from non-immunized mice (Figure 2C,D).

### 3.4. Screening of Immortalized B Cells for Production of Secreted or Membrane-Bound Igs Reactive against gp350

B cell lines generated from HIS mice immunized with EB-VLP were screened for secretion of gp350-specific human IgGs. Some lines secreted gp350-specific IgGs at levels comparable to that of a rat-derived hybridoma secreting a monoclonal gp350-specific antibody (6GA in Figure 3A) as tested in an ELISA assay with the recombinant gp350 protein. The specificities of these antibodies were confirmed in a subsequent flow cytometry assay using PCI 1 cells stably expressing gp350 on their surface (*p* = 0.0041) (Figure 3B,C).

### 3.5. Neutralization Capacity of IgGs Secreted by Immortalized B Cell Lines

Finally, we investigated whether supernatants of immortalized B cell lines produced gB- and gp350-specific IgGs that could neutralize HCMV and EBV infection in vitro, respectively. For HCMV, we used a neutralization assay, which is based on the TB40-GLuc HCMV strain used to infect MRC-5 fibroblasts [25,26]. Two out of six supernatants (LCL 2 and LCL 4) containing gB-specific IgGs revealed a neutralizing activity of up to 50% (Figure 4A,B). Similarly, we performed neutralization assays for EBV, where we analyzed the inhibition of infection in a flow cytometry-based assay. In fact, two out of four supernatants containing gp350-specific IgGs showed up to 70% inhibition of infection as compared to control supernatants that revealed no detectable neutralizing activities (3G5; *p* = 0.0569 and 1D9; *p* = 0.0374) (Figure 4C,D). Titration of the supernatants revealed the highest neutralizing capacity for supernatant for LCL 1D9 and 3G5 (internal name) (Figure 4E).

In summary, we show here that humanized NRG mice immunized with either iDCgB or EB-VLP produce virus-specific human IgG antibodies in vivo. EBV immortalization of splenocytes was successfully used to generate class-switched permanent LCL lines, which secrete antigen-specific IgGs, some of which have an in vitro neutralization capacity against HCMV and EBV. As a future outlook, the sequences of the characterized IgGs can be retrieved for the production of recombinant IgGs.

## 4. Discussion

We showed here that humanized NRG mice immunized with either iDCgB or EB-VLP produced antiviral human IgG antibodies in vivo. A cell culture system was developed by infecting splenocytes with EBV to generate immortalized B cell lines. Some of the expanded LCLs secreted IgGs with specific reactivity against CMV-gB or EBV-gp350. Ultimately, some of the screened LCLs secreted IgGs capable of neutralizing HCMV or EBV in vitro.

Humanized mice have become important model systems to study human-specific infections in vivo, as well as to investigate human immune responses [27]. Despite the success of humanized mouse models to follow human innate or T cell responses against immunizations or infections, B-cell responses have been considered absent or weak. In detail, humanized show an immature B cell development, with limited class-switch and antibody affinity maturation. In the past, several studies successfully used transgenic mouse strains with knock-ins of thymic-stromal-cell derived lymphopoietin (TSLP), human IL-6 and human leukocyte antigen (HLA) molecules, which improved B cell class switch and IgG serum levels significantly [28,29,30,31,32]. In another study, Spits and colleagues used humanized mice immunized with tetanus toxoid and an HBV vaccine and immortalized CD19^+^CD27^+^ B cells with human B cell lymphoma (BCL)-6 and BCL-XL genes in viral vectors, thereby obtaining permanent B cell lines that secreted antigen-specific IgM [33]. Using different immunization strategies such as lentiviral-modified dendritic cells or vectors, we showed that IgG immune responses against HCMV antigens pp65 or gB were obtained and consistently observed in humanized NRG mice [15,16].

We recently demonstrated that IgGs could be directly cloned from single cell sorted B cells [15], with the caveat that the cells used are destroyed for the purpose, an alternative technique to obtain monoclonal antibodies, could rely on the immortalization of cell lines for eventual downstream analyses. In addition, many biotechnological procedures require the use of a renewable biological material such as replicating immortalized cells. Thus, we successfully established an alternative protocol based on the immortalization of splenocytes from immunized mice with EBV to generate permanent IgG-secreting LCLs. We were able to demonstrate that some of these LCLs produced IgGs reactive against gB or gp350 and that some of them showed antiviral neutralization capacities. However, it must be stated that these neutralization capacities were, in general, lower than that of monoclonal antibodies that had been generated by immunizations of immunocompetent animals. Therefore, it is tempting to speculate that in HIS mice, the *lege artis* maturation of human B cells, which is a prerequisite for high-affinity antibodies, is not optimal. A putative confounding issue in our assays is that plasma and B cells obtained from immunized mice showed higher levels of IgG than B cells derived from non-immunized mice. Despite this correlative bias that may have resulted to some extent to higher IgG levels than our used control, IgGs derived from the transformed B cells derived from immunized animals showed reactivity against the cognate antigen to higher levels. Ig purification or normalization towards total Ig would improve comparability of immunized and non-immunized groups.

From a clinical perspective, the use of monoclonal antibodies to combat viral pathogens is being evaluated, for example, as a therapeutic option against HIV infections [34] and other viral infections [35,36]. For herpesviruses, such as HCMV and EBV, monoclonal antibody therapies have been explored in pre-clinical studies and clinical trials [37,38], however, none have reached routine clinical use. Furthermore, pooled and virus-specific antibody formulation are currently in clinical use, but full protection for these antibody pools has not been shown so far. In view of the lack of an approved vaccine for HCMV and EBV, monoclonal antibodies represent a potent treatment option to combat herpesvirus reactivation in immune-comprised patients.

Further analysis, such as cloning and screening of antibodies and subsequent potency testing in vitro and in vivo is necessary for a clinical translation, which were out of the scope of this current work and represents a limitation of this study. Additionally, this strategy can be used to immortalize B cells derived from several other tissues derived from humanized mice, such as bone marrow or lymph nodes. At this point, it would be also important to consider immunization, also a combined immunization with iDCs and VLPs, of humanized mouse models with an optimized B cell development as described above, to improve antibody affinity and neutralization capacity. A limitation of EBV immortalization is the variability of generated lines in vitro and that the lines are from an oligoclonal origin and not monoclonal. Therefore, additional improvements are needed to optimize and standardize the immortalization protocol towards the generation of single clones for instance by single cell sorting or telomere gene transfer.

Nevertheless, this study indicates the importance of humanized mice and B cell immortalization in order to generate novel and potent antibodies for clinical usage. Most likely, in a clinical setting, a combination of different antibodies targeting different immune-dominant antigens of both viruses would be the ideal clinical strategy in order to bypass viral mutational escape and immune evasion, which, in fact, has been a problem for clinical antibody therapy for infectious diseases.

In conclusion, our work demonstrates that (i) iDCgB and EB-VLPs induce antigen-specific immune responses in a pre-clinical humanized mouse model, (ii) EBV immortalized in vivo primed B cells derived from humanized mice secrete HCMV and EBV specific IgG`s with neutralization capacity, and (iii) humanized mice and EBV immortalization enable a novel technology platform for the production of fully human antibodies.

## 5. Conclusions

This is a proof-of-concept report demonstrating that the immunizations of humanized NRG mice with a DC-based vaccine expressing HCMV gB and EBV virus-like particles lead to induction of antigen-specific B cell responses and the production of fully human IgG antibodies. Immortalization of splenocytes recovered from these mice with EBV leads to the generation of permanent LCLs lines, which produced at low levels antigen-specific antibodies, some of which showed neutralizing capacity against HCMV and EBV in vitro. In conclusion, we present a novel platform to obtain antigen-specific fully human IgG antibodies from humanized mice.

## 6. Patents

The induced dendritic cell technology is described in the patent R. Stripecke, G. Salguero, A. Daenthasanmak, A. Ganser. “INDUCED DENDRITIC CELLS AND USES THEREOF” (PCT/EP2013/052485). Priority date 7 February 2013; Published 14 August 2014. International Publication Number WO 2014/122035A3. US application No: US14/820,645 granted on 30 April 2019, active. European patent No: EP2953644A2; EP2953644B1 granted on 18 June 2020, active. Japan patents Nr. 6866988 certificate of patent issued 27 April 2021, active. 

## Figures and Tables

**Figure 1 cells-13-00020-f001:**
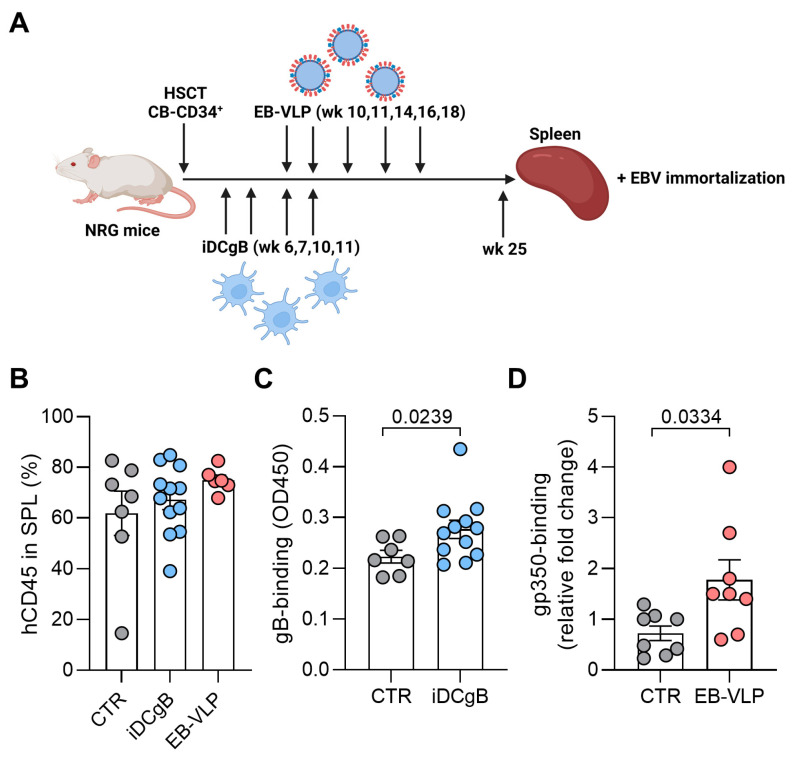
Fully humanized mice immunized with either iDCgB or EB-VLPs produce human IgGs reactive against gB and gp350, respectively. (**A**) Scheme of immunization of humanized mice and analyses. NRG mice were transplanted with CB-derived CD34^+^ stem cells. Afterwards, the animals were immunized either four times with iDCgB (autologous to the CB source) or five times with EB-VLPs. 25 weeks after HCT, the mice were sacrificed. Blood and spleen samples were harvested. (**B**) Reconstitution of humanized mice (CTR = grey; iDCgB = blue and EB-VLP = red) depicted in percentage (%) of human lymphocytes (CD45) in blood after 25 wks. Each dot represents one experimental mouse. (**C**) Detection of HCMV-gB specific IgG iDC (OD450) in plasma of humanized mice measured by ELISA. CTR not immunized are shown in grey and iDCgB immunized in blue. Each dot represents one mouse. (**D**) EB-VLPs immunized mice showed gp350-specific human IgG antibodies. The sera from immunized and control mice were used for an ELISA to detect gp350-specific human IgG antibodies. ELISA was performed with undiluted mice plasma obtained from CTR mice (grey) or from mice immunized with EB-VLPs (red). Standard deviation of the mean is indicated and students *t*-test with Welsh`s correction was used for statistical analysis as indicated.

**Figure 2 cells-13-00020-f002:**
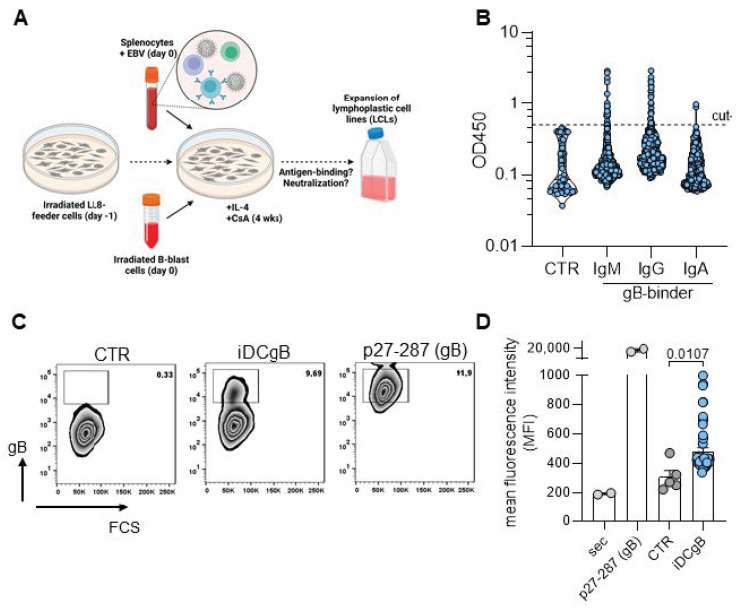
EBV immortalized B cell lines produce HCMV-gB-specific antibodies. (**A**) Schematic representation of the in vitro production of humanized mice-derived LCLs. (**B**) Screening example of lines derived from iDCgB immunized mice by detection of Igs in the cell supernatants binding to HCMV-gB by ELISA. Antibody subtype (IgM, IgG and IgA) is indicated and each dot represents results obtained for one cell line. Cut-off is indicated and was at OD 0.5 and based on IgG background signal obtained from LCL supernatants generated from non-immunized humanized mice (CTR). (**C**) Gating example of a flow cytometry based screening of B cell lines derived from iDCgB-immunized mice to detect gB-specific IgG. Supernatant was incubated with 293T cells expressing HCMV-gB on the cell surface and a specific binding was detected with a secondary antibody detecting human IgG. As positive control we used a gB specific primary antibody (p27-287). (**D**) Mean fluorescence intensity (MFI) quantification of all lines screened by flow cytometry. Sec: Secondary antibody used for detection; p27-287 (gB): Control primary antibody and secondary antibody used for detection; CTR: Supernatants from LCLs derived from non-immunized mice and stained with primary and secondary Abs; iDCgB: Supernatants from LCLs derived from iDCgB immunized mice and stained with primary and secondary Abs. Standard deviation of the mean is indicated and students *t*-test with Welsh`s correction was used for statistical analysis as indicated.

**Figure 3 cells-13-00020-f003:**
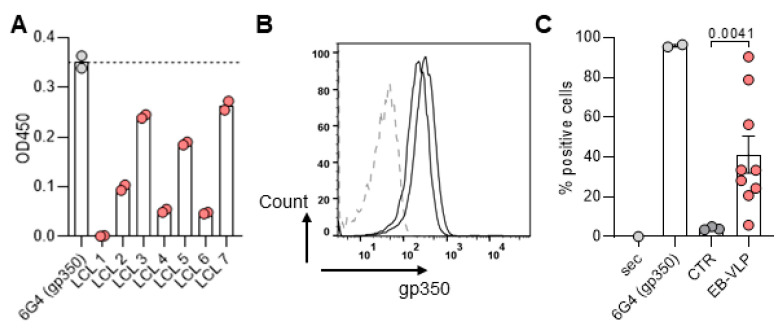
EBV immortalized B cells expressed EBV gp350-specific antibodies. (**A**) Supernatants LCLs generated from splenocytes from EB-VLPs immunized mice were tested by ELISA against recombinant gp350 protein and a few secreted gp350-specific human IgGs. 6G4 is a monoclonal antibody against gp350. (**B**) The supernatants of two representative lines were tested by flow cytometry for the presence of antibodies binding to PCI-1 cells stably expressing gp350 on the surface. The cells were stained with LCL supernatant (2 representative examples, black lines). Parental PCI-1 cells were used as negative control (dashed line). (**C**) Quantification of B cell in percentage of positive cells (%). Sec: Secondary antibody used for detection; 6G4 monoclonal antibody against gp350: Control primary antibody and secondary antibody used for detection; CTR: Supernatants from LCLs derived from non-immunized mice and stained with primary and secondary Abs; EBV-VLP: Supernatants from LCLs derived from VLP immunized mice and stained with primary and secondary Abs. Standard deviation of the mean is indicated and students *t*-test with Welsh`s correction was used for statistical analysis as indicated.

**Figure 4 cells-13-00020-f004:**
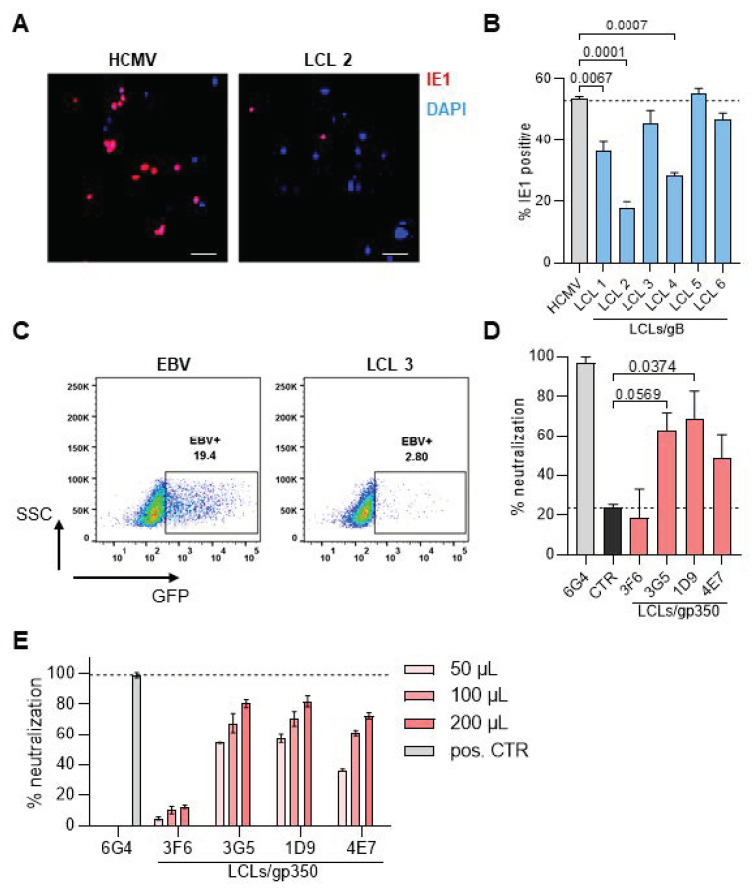
Supernatants of immortalized B cell lines neutralize HCMV and EBV infections in vitro. (**A**) The HCMV in vitro neutralization assay was performed with MRC-5 cells infected with the HCMV strain TB40-GLuc. DAPI (blue) was used to quantify the total cells and the fraction of cells expressing HCMV-IE1 (magenta) was used to quantify infection. Microscope magnification is 20× and scale bar represents 50 µm. (**B**) Percentage of IE1-positive cells within all cells when supernatants of different LCLs targeted against gB were added to the assay. (**C**) The EBV in vitro neutralization assay was performed with Raji cells infected with the EBV expressing GFP (MOI 0.2). 3 days after infection, the infected cells were quantified by flow cytometry through GFP expression (left panel). (**D**) Quantified values of EBV neutralization performed with supernatants obtained from different LCLs LCLs targeted against gp350. The 6G4 gp350-specific rat monoclonal antibody was used as positive control (grey). Control LCLs are shown in black. A gp350 negative LCL was used as ref. control. The percentage of inhibition was calculated normalizing to EBV infection. (**E**) The same experiment was done titrating the amount of supernatants used for neutralization. For all experiments *n* = 2 was performed and standard deviation of the mean is depicted in the graph. Statistics were performed using One-way-ANOVA and *p*-values are shown.

## Data Availability

The data presented in this study are available on request from the corresponding author. All generated data is represented within this manuscript.

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
