# Peer review of "Fully Human Herpesvirus-Specific Neutralizing IgG Antibodies Generated by EBV Immortalization of Splenocytes-Derived from Immunized Humanized Mice"

_cells, 2023, doi:10.3390/cells13010020_

Round 1
Reviewer 1 Report
Comments and Suggestions for Authors
Humanized mice (hu-mice) with a human immune system are an excellent model to study human immune responses in an experimental animal model. Human B cells develop robustly in this model and this group has previously developed a novel immunization method that improves antibody responses following vaccination.
The scope of this current study was to establish whether immunized hu-mice can be used to develop human B cell lines that produce virus-specific neutralizing monoclonal antibodies. Monoclonal antibodies can be used prophylactically or therapeutically to prevent or cure infections, respectively, and represent a desirable treatment for emerging infections.
This study aimed to test the following concepts: whether vaccination of hu-mice with either autologous engineered dendritic cells expressing the human cytomegalovirus gB envelope protein (HCMV-gB) or with Epstein-Barr virus-like particles (EB-VLP) promote the development of virus-specific anti-sera, whether human B cells from immunized hu-mice can be immortalized with EBV, and whether immortalized B cell lines from vaccinated hu-mice produce virus-specific monoclonal antibodies of neutralizing capacity.
Overall, the manuscript is well written and the study is straightforward. The results of the study are promising: the sera of immunized mice exhibit increased reactivity toward the respective viral antigens, B cells from these animals can be immortalized via EBV infection and, most importantly, some of the B cell lines appear to produce monoclonal antibodies that are not only reactive toward the viral antigens, but that can also neutralize the viruses themselves.
The study however, is a bit preliminary in its execution and scope.
For instance, the mouse sera and the supernatant of the B cell lines were tested (in ELISA, flow cytometry, and neutralization assays) without apparent normalization. The background of these assays is heavily influenced by total Ig amounts. Thus, if the supernatants were not first normalized for total IgG or IgA, it is not possible to compare the results of the assays.
It is the opinion of this reviewer that cloning and sequencing of the antibodies from the reactive B cell lines is within the scope of this work: determining whether the neutralizing antibodies use heavy and light chain genes that have been previously described in humans and establishing whether these recombinant antibodies can prevent or reduce infection of hu-mice would provide the proper completion of this study.
Minor comments:
Figure 1 legend: legend of panel E is missing
Reviewer 2 Report
Comments and Suggestions for Authors
The authors present a strategy for producing human IgG antibodies using human immune system mice produced through cord blood CD34+ cell injection into NRG mice. They demonstrate two strategies to induce class-switched antibody responses in the mice that involved multiple immunizations with either iDCs transduced with the HCMV-gB antigen or EBV-like viral particles. After immunization, splenocytes are harvested and the human B cells undergo EBV immortalization to produce antibody secreting lymphoblastoid cell lines (LCLs). Some cell lines are then demonstrated to bind to cells expressing the vaccine antigen or to neutralize infection by the virus corresponding to the vaccine antigen.
Overall, the paper is very well written and the data presented clearly and professionally. The ability to isolate antigen specific human IgG expressing clones from humanized mice is an attractive and feasible methodology of significant interest. However, some improvements are required prior to publication:
· Please give information as to the origin of the recombinant EBV GFP virus in the methods section 2.10
· Figure 1E is misplaced. It should be panel A in Figure 2.
· It is unclear in the current Fig 2A how cut-off was determined. The OD450 for non-immunized mice is how this should have been derived and should be shown.
· The positive control antibody used to determine MFI in Fig 2C should also be shown for % positive staining in 2B.
· In section 3.4 it is stated that LCL 1D9 revealed the “highest neutralization” however this is not specifically shown/quantified statistically and is quite similar to 3G5.
· Experiments in Figure 4B, D and E appear to be single technical replicates for each LCL. Additional technical replicates, standard deviations and statistics are required to compare binding and neutralizing capacity of LCL supernatants to each other and to controls.
· Figure 4 D legend is confusing. Refers to "(grey)" when both positive and negative control bars are grey.
Author Response
Reviewer 2:
The authors present a strategy for producing human IgG antibodies using human immune system mice produced through cord blood CD34+ cell injection into NRG mice. They demonstrate two strategies to induce class-switched antibody responses in the mice that involved multiple immunizations with either iDCs transduced with the HCMV-gB antigen or EBV-like viral particles. After immunization, splenocytes are harvested and the human B cells undergo EBV immortalization to produce antibody secreting lymphoblastoid cell lines (LCLs). Some cell lines are then demonstrated to bind to cells expressing the vaccine antigen or to neutralize infection by the virus corresponding to the vaccine antigen. Overall, the paper is very well written and the data presented clearly and professionally. The ability to isolate antigen specific human IgG expressing clones from humanized mice is an attractive and feasible methodology of significant interest. However, some improvements are required prior to publication:
Response: We thank the reviewer for the supportive comments.
- Please give information as to the origin of the recombinant EBV GFP virus in the methods section 2.10
Response: The recombinant GFP-bearing EBV has been described in: Delecluse HJ, Hilsendegen T, Pich D, Zeidler R, Hammerschmidt W. Propagation and recovery of intact, infectious Epstein-Barr virus from prokaryotic to human cells. Proc Natl Acad Sci U S A. 1998 Jul 7;95(14):8245-50. (see lines 208-209, reference 25)
- Figure 1E is misplaced. It should be panel A in Figure 2.
Response: Thanks for the suggestion. Figure 1E appears now as new Figure 2A.
- It is unclear in the current Fig 2A how cut-off was determined. The OD450 for non-immunized mice is how this should have been derived and should be shown.
Response: We thank the reviewer for this comment, which was also clarified for Reviewer 1. The cut-off was determined based on the background signal of immortalized B cells derived from non-immunized humanized mice. Therefore we added the data obtained from non-immunized into the figure and explained the cut-off within the corresponding figure legend.
- The positive control antibody used to determine MFI in Fig 2C should also be shown for % positive staining in 2B.
Response: We thank the reviewer for this comment. As suggested we added the positive control to the figure – now 2C.
- In section 3.4 it is stated that LCL 1D9 revealed the “highest neutralization” however this is not specifically shown/quantified statistically and is quite similar to 3G5.
Response: We thank the reviewer for this comment. We agree with the reviewer that both LCLs showed similar neutralization. We therefore amended the manuscript accordingly (lines 317-320). We also added statistics to Figure 4D, indicating the lower p-values for 1D9 compared to 3G5 in comparison to control.
- Experiments in Figure 4B, D and E appear to be single technical replicates for each LCL. Additional technical replicates, standard deviations and statistics are required to compare binding and neutralizing capacity of LCL supernatants to each other and to controls.
Response: We thank the reviewer for this comment. We completed the graphs with the standard deviations and statistics.
- Figure 4 D legend is confusing. Refers to "(grey)" when both positive and negative control bars are grey.
Response: We apologize for this confusing legend. The color coding of the graph and the figure legend were improved.
Reviewer 3 Report
Comments and Suggestions for Authors
This paper attempts to create anti-viral antibodies by transplanting human blood into severely immunodeficient mice. The experimental results demonstrate the induction of antibody production by iDCgB and EBV-VLPs and the establishment of a long-term antibody production system through LCL establishment.
However, high-quality antibodies that strongly suppress viral infection have not been obtained. Additional experiments are not necessary, but problems with introduction, the experimental method, and discussion part need to be noted.
In introduction
1. Mice are not infected with EBV or human CMV. Therefore, only transplanted human cells are infected with these viruses. However, VLP was used in the experiment. In the introduction, it is necessary to explain the advantages of using VLPs as antigens. Of course, the explanation of what VLP is missing. Similarly, it needs the explanation of use antigen-expressing dendritic cells instead of CMV.
2. The authors used NRG to produce anti-virus human antibodies, but there are several different types of immunodeficient mice such as NOG, NSG, and NRG are existed. It is necessary to explain in the introduction the advantages of NRG compared to other mice.
In methods
1. There is no mention of where the MRC-5 was purchased from.
2. PCI-1 and LL8 were obtained from two researchers. To unify the description of the paper, the full name of the researcher should be included.
3. State the MOI of the lentivirus.
4. The Hammarshumit lab had multiple generations of cells to create VLPs. Please describe which generation the cells used correspond to and which genes are missing. Also, the virus strain should be listed. Also write Professor Hammershimidt's full name.
5. Some full spellings are missing. PBS (phosphate-buffered saline), LCL (Lymphoblastoid cell line), VLP (Virus-like particle) etc.
In discussion
1. The data in Figures 1C and D and Figure 2A show that the immune response is weaker than in normal mice. In NOG mice, the efficiency of dendritic cell generation is very low, and the MHC does not match the transplanted cells. Additionally, there is a loss of lymph nodes. Therefore, NOG mice have a low antigen-specific immune response. Administration of iDCgB enhances lymph node formation, but the results in Figure 4B show that antibodies that completely inhibit CMV infection have not been obtained. A similar trend is also observed from the EBV antibody titers in Figure 4D. Therefore, problems not only with LCL cloning but also with the animal model itself are expected. Therefore, even if an immune response is detected, it is necessary to clearly state that future experiments using improved mice are required. Furthermore, it is difficult to obtain high-quality antibodies when EBV VLPs are administered alone because there are no dendritic cells. The combination of EBV and iDC also should be described.
2. If there are B cells that produce antibodies with extremely high neutralizing activity, it is expected that the EBV added for immortalization will be neutralized by the antibodies in the culture supernatant and will not be able to infect the B cells. In fact, in Figure 3A, 4 out of 7 clones showed low binding to gp350 (Figure 3A). Furthermore, LCL with neutralizing antibody activity reaching 100% was not obtained (Fig. 4D, E). To obtain high-quality EBV antibodies, it is necessary to state that LCL alone is technically insufficient due to its principle. Therefore, other techniques such as telomere gene transfer should also be mentioned in the discussion.
3. Once an antibody with neutralizing activity against the virus is confirmed, high-titer antibodies can be obtained by identifying the gene sequence and incorporating it into an expression vector. A method for obtaining antibodies without going through the immortalization step should be described in the discussion.
Comments on the Quality of English LanguageThis paper frequently misses full spellings. For this reason, English proofreading is required.
Author Response
Reviewer 3:
This paper attempts to create anti-viral antibodies by transplanting human blood into severely immunodeficient mice. The experimental results demonstrate the induction of antibody production by iDCgB and EBV-VLPs and the establishment of a long-term antibody production system through LCL establishment.
However, high-quality antibodies that strongly suppress viral infection have not been obtained. Additional experiments are not necessary, but problems with introduction, the experimental method, and discussion part need to be noted.
Response: We thank the reviewer for the supportive comments. Please note that the mice were transplanted with isolated human CD34+ progenitor cells obtained from cord blood. We modified and revised introduction, methods and discussion parts.
In introduction
- Mice are not infected with EBV or human CMV. Therefore, only transplanted human cells are infected with these viruses. However, VLP was used in the experiment. In the introduction, it is necessary to explain the advantages of using VLPs as antigens. Of course, the explanation of what VLP is missing. Similarly, it needs the explanation of use antigen-expressing dendritic cells instead of CMV.
Response: There was a misunderstanding by the reviewer. The cord blood cells were not infected with viruses. The reason why we used B cells derived from humanized mice immunized with iDCgB instead of B cells derived from humanized mice infected with HCMV, is because the later produced inferior antibody responses (see Theobald et al., Plos Pathogens 2021). In the case of EBV, the mice develop tumors and therefore a B cell response is mainly influenced by tumor formation and not antigen-specific driven as from an immunization. This argument holds also true for HCMV, as these mice develop a broad immune response compared to only single-antigen immunized mice. Therefore, we added a paragraph to our introduction (lines 85-89).
- The authors used NRG to produce anti-virus human antibodies, but there are several different types of immunodeficient mice such as NOG, NSG, and NRG are existed. It is necessary to explain in the introduction the advantages of NRG compared to other mice.
Response: We thank the reviewer for this comment and therefore we added a paragraph to the introduction (lines 74-77) explaining the advantage of NRG mice to other models.
In methods
- There is no mention of where the MRC-5 was purchased from.
Response: MRC-5 cell were purchased from ATCC (line 105).
- PCI-1 and LL8 were obtained from two researchers. To unify the description of the paper, the full name of the researcher should be included.
Response: We modified the text according to the reviewer’s suggestion (lines 110 and 112).
- State the MOI of the lentivirus.
Response: We thank the reviewer for this comment. The concentration of both lentiviral vectors are now mentioned (lines 132-135). Of note, we use mg/ml p24 equivalent measured by ELISA as measurement for transduction (see reference 15)
- The Hammarshumit lab had multiple generations of cells to create VLPs. Please describe which generation the cells used correspond to and which genes are missing. Also, the virus strain should be listed. Also write Professor Hammershimidt's full name.
Response: We thank the reviewer for this comment. We added a paragraph to our current Material and Method (lines 141-145). Please also see Bouvet…Hammerschmidt et al. (Ref. 21) for a detailed description. The name was modified as suggested by the reviewer.
- Some full spellings are missing. PBS (phosphate-buffered saline), LCL (Lymphoblastoid cell line), VLP (Virus-like particle) etc.
Response: We included the full spelling prior to the abbreviations.
In discussion
- The data in Figures 1C and D and Figure 2A show that the immune response is weaker than in normal mice. In NOG mice, the efficiency of dendritic cell generation is very low, and the MHC does not match the transplanted cells. Additionally, there is a loss of lymph nodes. Therefore, NOG mice have a low antigen-specific immune response. Administration of iDCgB enhances lymph node formation, but the results in Figure 4B show that antibodies that completely inhibit CMV infection have not been obtained. A similar trend is also observed from the EBV antibody titers in Figure 4D. Therefore, problems not only with LCL cloning but also with the animal model itself are expected. Therefore, even if an immune response is detected, it is necessary to clearly state that future experiments using improved mice are required. Furthermore, it is difficult to obtain high-quality antibodies when EBV VLPs are administered alone because there are no dendritic cells. The combination of EBV and iDC also should be described.
Response: We thank the reviewer for the valuable comments. We agree that humanized mice (we used NRG mice and not NOG mice) show low antigen-specific immune responses, influenced by low numbers of lymph nodes and MHC mismatch. Just to clarify, the dendritic cells were autologous to the CD34+ cells and therefore fully HLA matched (see revision in legend of Fig. 1, line 258). However, we want to emphasize the point that immunized mice in Figure 1C and 1D showed significant higher levels of IgGs reactive against gB and gp350 in the plasma compared to non-immunized mice. This clearly shows that immunization leads to an improved immune response. We agree to the reviewer that we neither supernatants completely inhibiting HCMV or EBV infection in vitro. We also agree that immunization of improved humanized mouse models would potentially improve antibody neutralization capacity. Therefore, we added a paragraph to your discussion (line 386-389), as suggested by the reviewer. Further, we agree to the reviewer’s idea to use our iDC technology for the presentation of EBV antigen in humanized mice to improve immune responses. We tried this strategy, but could not demonstrate that DCs transduced with a LV expressing gp350 expressed the antigen. Therefore, we used VLPs, that were immunogenic to induce B cell immune responses.
- If there are B cells that produce antibodies with extremely high neutralizing activity, it is expected that the EBV added for immortalization will be neutralized by the antibodies in the culture supernatant and will not be able to infect the B cells. In fact, in Figure 3A, 4 out of 7 clones showed low binding to gp350 (Figure 3A). Furthermore, LCL with neutralizing antibody activity reaching 100% was not obtained (Fig. 4D, E). To obtain high-quality EBV antibodies, it is necessary to state that LCL alone is technically insufficient due to its principle. Therefore, other techniques such as telomere gene transfer should also be mentioned in the discussion.
We agree with the interesting insight of the reviewer that, in principle, B cells producing neutralizing antibodies against gp350 could have been protected from infection, and hence would not become immortalized. In fact, this may be a reason for not having obtained antibodies with very high neutralization capacities (and thus high affinities). However, we did obtain immortalized B cells producing anti-gp350 antibodies. In addition, we rather think that this was a result of a suboptimal maturation of human B cells in mice. This has been added to our current discussion (line 366-372)
The biological properties of antibodies are determined in immunized animal in vivo and include the route of immunization, the immunogen and adjuvant applied, as well as several crucial steps of B-cell development, including affinity maturation, somatic hypermutation, class switch etc. We agree that LCLs are probably not an ideal platform for Ig production, as their productivity is low. Alternative technologies including telomerase-immortalized B cells would be a suitable alternative, resulting in a higher amount, yet not quality, of the antibodies produced. We added a paragraph to your current discussion (line 391-393).
- Once an antibody with neutralizing activity against the virus is confirmed, high-titer antibodies can be obtained by identifying the gene sequence and incorporating it into an expression vector. A method for obtaining antibodies without going through the immortalization step should be described in the discussion.
Response: We thank the reviewer for this comment and agree that that single cell sorting as we showed before is a potential alternative method to obtain antibodies with neutralization capacity from humanized mice. We would like to mention at this point that we discussed and mentioned single cell sorting of B cells from humanized mice within the current manuscript (lines 360-363 and lines 382-384).
Reviewer 4 Report
Comments and Suggestions for Authors
In this article, Sebastian and colleagues discussed a method to generate fully human antibodies for therapeutic purposes using humanized mice. The authors demonstrate that B cells recovered from humanized mice can be immortalized with Epstein-Barr virus (EBV) using a feeder cell system. The immortalized B cells secrete human IgGs with neutralization capacities against human cytomegalovirus (HCMV) and Epstein-Barr virus (EBV). This technology can be further explored to generate antibodies against emerging infections for diagnostic or therapeutic purposes.
Some comments:
1: The article mentions that statistical analysis was performed using GraphPad Prism V7.0 software, but it does not provide any specific details about the statistical tests used or the results of the analysis.
2: The result and discussion part are not well written.
3: in line 50, which method did they use in production of SARS-CoV-2 antibody?
4: in line 150, I did not see the supplemental material information.
5: in figure 1C, compared to CTR, the immunization group also showed low OD. It made it unsuccessful immuration procedure, or did the author measure humanization efficiency?
6: Did the author test the affinity of the produced antibodies?
7: in figure 2A, how did the author set the cut-off to 0.5? what’s the role of IgA against infection of HCMV?
8: Did the antibodies get purified in neutralization assay? What’s the concentration did the author use?
Author Response
Reviewer 4:
In this article, Sebastian and colleagues discussed a method to generate fully human antibodies for therapeutic purposes using humanized mice. The authors demonstrate that B cells recovered from humanized mice can be immortalized with Epstein-Barr virus (EBV) using a feeder cell system. The immortalized B cells secrete human IgGs with neutralization capacities against human cytomegalovirus (HCMV) and Epstein-Barr virus (EBV). This technology can be further explored to generate antibodies against emerging infections for diagnostic or therapeutic purposes.
Response: We thank the reviewer for the supportive comments.
Some comments:
1: The article mentions that statistical analysis was performed using GraphPad Prism V7.0 software, but it does not provide any specific details about the statistical tests used or the results of the analysis.
Response: We thank the reviewer for his comments. In the current manuscript we mentioned each statistical method within the respective figure legend. In order to clarify we added a paragraph to the statistical method description.
2: The result and discussion part are not well written.
Response: We apologize to the reviewer. The results section was modified at several points (also based on the comments of the additional reviewers) (Figure 1 and 2). We additionally revised the discussion at several parts (lines 346-350, 360-372, 382-393).
3: in line 50, which method did they use in production of SARS-CoV-2 antibody?
Response: We thank the reviewer for this comment. We cited the clinical study of the monoclonal antibody showing efficacy in a human clinical trial. However, the used antibody against SARS-CoV-2 was originally isolated from a human donor using EBV immortalization (same technology as represented here) of human memory B cells (PMID: 32422645).
4: in line 150, I did not see the supplemental material information.
Response: We apologise as during our internal reviews we decided to show all the antibodies in the main text. Therefore, no supplemental information was needed.
5: in figure 1C, compared to CTR, the immunization group also showed low OD. It made it unsuccessful immuration procedure, or did the author measure humanization efficiency?
Response: The non-vaccinated and vaccinated mice showed similar humanization levels as followed by the frequency of huCD45+ cells. But the vaccine Although the CTR group showed a background signal, the immunized group showed a significantly higher signal. As addressed for Reviewer 1, in previous studies, we demonstrated that humanized mice immunized with iDCgB showed significantly higher absolute numbers of class-switched B cells expressing IgG, IgM and IgA in spleen and bone marrow (Theobald et al, Plos Pathogens, 2021).
6: Did the author test the affinity of the produced antibodies?
Response: We did not test the affinities. At this point we would like to show the proof-of-principle of the technology per se.
7: in figure 2A, how did the author set the cut-off to 0.5? what’s the role of IgA against infection of HCMV?
Response: We thank the reviewer for this comment. The cut-off was set to 0.5 as this was the background signal obtained by our reference LCL supernatants, which were generated of splenocytes derived from non-immunized mice. The data is now depicted in the new Figure 2B, as also requested by reviewer 2.
The role of role of IgA against HCMV infection is not well understood. It is known that humans develop IgA antibodies against HCMV proteins (PMID: 33283275, PMID: 8395539) for instance against gB, the target protein in our study. Because of the high presence in the mother’s milk, it is postulated that IgA have an important role in protecting neonates from HCMV infection. Further, it is well known that IgA plays an important role in the mucosa and therefore in the protection against respiratory viruses, such as HCMV. Therefore, we aimed to evaluate if our technology also elicits IgA antibodies against gB. However, as written in our study the IgA response was very minor to not existing.
8: Did the antibodies get purified in neutralization assay? What’s the concentration did the author use?
Response: As mentioned above, we did not purify the supernatants throughout the manuscript and therefore did not use specific antibody concentrations. For the neutralization assay in-particular, we tested the LCL supernatants for their neutralization capacity. For HCMV we did not perform dilutions of the supernatants. In our first EBV neutralization assay (Figure 4C and 4D) we used pure supernatants and for the second neutralization assay (Figure 4E) we diluted the supernatants to show a dose dependency.
Round 2
Reviewer 1 Report
Comments and Suggestions for Authors
The authors were only partially responsive to the critiques.
The figure they provide in their response showing that immunized mice harbor higher amounts of sera Ig is exactly what the comment was about. Sera (or supernatants) with higher Ig levels result in higher signal (or background). That is why it is important to normalize the amount of total Ig. This means that supernatants should be differentially diluted before an assay to input in the assay the same amount of Ig.
Minimally, the Discussion should include the absence of Ig normalization as a caveat to these assays.
In regard to Ig gene sequencing, this methodology is quite routine nowadays and would take very little time to sequence the genes of the cell lines expressing neutralizing antibodies. Nevertheless, this reviewer accepts the point that the authors wish to leave that analysis for a followup publication.
Author Response
The authors were only partially responsive to the critiques.
The figure they provide in their response showing that immunized mice harbor higher amounts of sera Ig is exactly what the comment was about. Sera (or supernatants) with higher Ig levels result in higher signal (or background). That is why it is important to normalize the amount of total Ig. This means that supernatants should be differentially diluted before an assay to input in the assay the same amount of Ig.
Minimally, the Discussion should include the absence of Ig normalization as a caveat to these assays.
In regard to Ig gene sequencing, this methodology is quite routine nowadays and would take very little time to sequence the genes of the cell lines expressing neutralizing antibodies. Nevertheless, this reviewer accepts the point that the authors wish to leave that analysis for a followup publication.
We thank the reviewer for his comments. According to the reviewers comments we added a paragraph to the discussion (lines 372-379), stating that Ig normalization is a limitation to our read-outs.
Reviewer 3 Report
Comments and Suggestions for Authors
The authors have prepared thorough answers to the reviewers' questions.
Author Response
We thank the reviewer again evaluating the manuscript.
Reviewer 4 Report
Comments and Suggestions for Authors
good to go after this revision.
Author Response
We thank the reviewer for the comments.